# Control of Protein Homeostasis in the Early Secretory Pathway: Current Status and Challenges

**DOI:** 10.3390/cells8111347

**Published:** 2019-10-29

**Authors:** Daria Sicari, Aeid Igbaria, Eric Chevet

**Affiliations:** 1Proteostasis & Cancer Team INSERM U1242 « Chemistry, Oncogenesis Stress Signaling », Université de Rennes, CEDEX, 35042 Rennes, France; aeid.igbaria@gmail.com; 2Centre de Lutte contre le Cancer Eugène Marquis, CEDEX, 35042 Rennes, France

**Keywords:** ER stress, Golgi stress, ERAD, EGAD, protein quality control

## Abstract

Discrimination between properly folded proteins and those that do not reach this state is necessary for cells to achieve functionality. Eukaryotic cells have evolved several mechanisms to ensure secretory protein quality control, which allows efficiency and fidelity in protein production. Among the actors involved in such process, both endoplasmic reticulum (ER) and the Golgi complex play prominent roles in protein synthesis, biogenesis and secretion. ER and Golgi functions ensure that only properly folded proteins are allowed to flow through the secretory pathway while improperly folded proteins have to be eliminated to not impinge on cellular functions. Thus, complex quality control and degradation machineries are crucial to prevent the toxic accumulation of improperly folded proteins. However, in some instances, improperly folded proteins can escape the quality control systems thereby contributing to several human diseases. Herein, we summarize how the early secretory pathways copes with the accumulation of improperly folded proteins, and how insufficient handling can cause the development of several human diseases. Finally, we detail the genetic and pharmacologic approaches that could be used as potential therapeutic tools to treat these diseases.

## 1. Introduction

Secretory and transmembrane proteins are synthesized in the endoplasmic reticulum (ER) and then transported to the Golgi compartment for additional maturation. Acquisition of a correct conformation is a critical step for proteins functionality and targeting to their final destination. This is achieved mainly by the aid of numerous ER and Golgi apparatus resident chaperones as well as folding modifying enzymes [1].

ER protein folding is an error prone process especially because of the huge number of possible conformations that proteins can acquire during folding. To ensure quality control through this process, both the ER and the Golgi exhibit stringent and complex proteostasis network to coordinate their functions. ER-dependent protein quality control imposes a collaboration between protein synthesis and productive folding, ER-associated degradation (ERAD) [2] and ER homeostasis control (unfolded protein response; UPR) [3]. In addition, Golgi-dependent quality control mechanisms are necessary to monitor polypeptides maturation and post-translational modifications [4]. For instance, modifications such as N-linked glycosylation, which is critical for protein recognition and integral maturation, is initiated in the ER and completed in the Golgi. Several environmental and chemical conditions can alter the functionality of the ER and Golgi complex causing the accumulation of improperly folded proteins, which impinge on cellular proteostasis. To prevent immature proteins production and accumulation, mammalian cells developed different strategical quality checkpoints along the secretory pathway starting from their synthesis until their final destination. These different checkpoints are important to prevent the aberrant secretion of improperly folded proteins, which could in turn become proteotoxic and cause several human diseases (e.g., degenerative and proliferative disorders) [5].

## 2. Maintenance of Early Secretory Pathway Homeostasis and Impact on Protein Secretion

Secretory proteins and most transmembrane proteins first enter the ER, the first compartment of the secretory pathway, in order to acquire fundamental modifications that are necessary for their maturation. These events include for instance N-glycosylation, disulfide bond formation or proline isomerization [6]. During their journey in the early secretory pathway, secretory and transmembrane proteins pass through the ER, which they leave through ER exit sites (ERES) into the ER-to-Golgi intermediate compartment (ERGIC) before they reach the Golgi complex. These processes are facilitated by chaperones and folding enzymes that assist protein folding in the ER and allow them to skip quality control checkpoints. Newly synthesized proteins progressively adopt conformations that are energetically more favorable, until the correctly folded ‘native’ conformation is reached [7].

### 2.1. Foldases, Chaperones and Quality Control Mechanisms

Once translated, proteins reach the ER lumen and due to the presence of numerous resident chaperones and modifying enzymes, can continue their travel along the secretory pathway till the complete maturation. In the other hand, immature proteins will be degraded (Figure 1).

Chaperones and enzymes in the ER have different mechanisms to help protein folding. Two representative ER-resident chaperones, the heat shock protein (HSP70) glucose-regulated protein 78/Bip (GRP78/BIP) and the HSP90 family member GRP94, share a similar mechanism. In particular, they bind the hydrophobic region of unfolded proteins and facilitate folding through conformational changes and ATP hydrolysis mediated by their ATPase domains [7]. Notably, a separate class of chaperones and co-chaperones that belongs to the HSP40 family is in charge of maintaining BiP’s ATPase-domain functionality. Among them ERdj3, which relies on a multi-protein complex, binds to BiP in its ATP bound form and is necessary for BiP to recruits its substrates [8]. On the other hand, the co-chaperone GRP170 binds BiP in ATP depleted conditions. Other molecules such as ERdj6/p58IPK and ERdj4 were shown to directly bind to unfolded proteins [9]. Moreover, ERdj4 was found together with ERdj5 to be associated with the ERAD complex, thus implicating their involvement in protein quality control and degradation rather than in protein folding. Finally, apart from the malfolded protein, ERdj4 and BiP were demonstrated necessary to bind the luminal domain of the ER stress receptor IRE1 thus inhibiting its activation in non-stress conditions [10]. BiP was also found in complex with numerous protein disulfide-isomerases and peptidyl-prolyl cis–trans-isomerases, these associations appear necessary for promoting folding of nascent polypeptide chain. As a matter of fact, in vitro experiments show that protein disulfide isomerase (PDI) and BiP cooperate in antibodies folding, in particular, it was suggested that BiP, by binding the polypeptides, allows PDI access to their cysteine [11]. In addition to this, BiP is also target post-translational modifications, which control its activity. For instance, BiP acetylation on lysine 585 was found to inhibit it’s binding on PERK, leading to activation of the unfolded protein response (UPR) [12,13]. Moreover, BiP, upon glucose starvation, became ADP-ribosylated and inhibited [14]. Glycoproteins maturation and sorting is mainly under the control of a subset of lectins such as ERGIC53 and the mannose-6 phosphate receptor in the Golgi. Diverse lectins in the ER show diverse glycans specificities. The most important canonical ER lectins are calnexin (CANX) and calreticulin (CALR), which dictate glycoprotein maturation and release in the secretory pathway [1]. This step includes the activity of different enzymes, such as glucosidase I or II that control trimming of two glucose residues [1]. As a result, proteins that have only one bound glucose residue will be recognized as a substrate for the CANX/CALR folding complex [15]. It was largely demonstrated that the protein disulfide isomerase ERp57 participates in this process by binding the extended P-domain of calnexin/calreticulin and accelerating glycoprotein folding [16]. Once the last glucose is trimmed, client proteins are released and bound by an UDP-glucose–glycoprotein glucosyltransferase (UGGT) [15]. The UGGT is a molecular chaperone, which, due to a large N-terminal region, can recognize diverse misfolded glycoprotein substrates and specifically add a glucose moiety to the Man9 containing glycan. Glc1-Man9 glycoprotein can rebind calnexin/calreticulin and undergo further folding cycles catalyzed by ERp57. If the protein is correctly folded, it can move to the other steps. If not, a new glucose residue is attached and the client protein enters again into the CANX/CALR cycle [15].

### 2.2. Post-Translational Modifications in the ER

Both N-glycosylation and O-mannosylation are pivotal for proteins maturation. N-linked glycosylation sites, usually composed of asparagine followed by any amino acid that is not proline, which is followed by serine/threonine sequences, are the sited in which high mannose oligosaccharide are attached [17]. Finally, O-mannosylation, which will occur on the Ser and Thr side-chain, will improve protein stability and solubility [18]. Nascent polypeptides initially interact with the two oligosaccharyl transferase (OST) complexes that transfer the core Glc_3_Man_9_GlcNAc_2_ (Glc is glucose, Man is mannose and GlcNAc is N-acetylglucosamine) glycan from lipid donor to the asparagine side chain. The O-mannosylation process involves a single mannose residue linked to Ser and Thr side-chain hydroxyls in the α-configuration and is crucial for protein quality control. Notably, it starts in the ER lumen, but can be further extended in the Golgi apparatus thanks to the activity of a mannosyltransferase, which catalyzes the transfer of mannoses from GDPα-d-mannose. As already mentioned, together with glycosylation others post-translational modifications (PTM) occur in the ER and are mediated by ER-resident enzymes. For example, cysteine thiols oxidation is mediated by protein disulfide isomerases (PDIs) [19]. This class of 20 proteins is strongly induced upon ER stress induction and is involved both in protein folding and degradation. Structurally, the members of this family show an ER retention motif and one or more, inactive and active, domains with a predicted thioredoxin-like structure. However, some of them like ERp29 and 27 only has inactive domains but both present chaperone activity. On the other hand, ERp19 and AGR2 exhibit a single active thioredoxin-like domain without demonstrated chaperone activity. The meaning of these differences between the member of the family is still unknown and should be of interest for further investigations. Another class of molecules involved in the folding process is the rotamases known as peptidyl-prolyl cis–trans isomerases (PPIs) [20]. These proteins catalyze the isomerization of Xaa-proline peptide bonds. This represents a rate-limiting step in the folding of several proteins, including antibodies, and it occurs when the maturation is almost complete. Mechanistically, it was proposed that the different ER-resident PPIs (cyclophilin B, FKBP13, FKBP23, and FKBP65) contain a chaperone-like substrate binding domain or use other ER proteins to bind client polypeptide [20]. As matter of fact, cyclophilin B does not contain this binding site but relies on a bigger complex capable to recruit proteins. Indeed, it was found associate with IgG in vivo and it was shown to slowly increase the isomerization step in vitro, leading to the correctly folded dimeric species. On the other hand, FKBP65 shows chaperone-like activity since it was demonstrated to bind collagen and FKBP23 acts as BiP ATPase activity inhibitor [20]. However, unlike to disulfide isomerization, it is not still clear whether PPIs mediated isomerization is important for degradation or/and folding, but further studies are necessary to clarify how PPIs bind client proteins and their functions in protein folding.

### 2.3. Proteostasis Maintenance in the Early Secretory Pathway—Signaling Aspects

Proteostasis maintenance is necessary to prevent production of mutated proteins and their accumulation, to retain uncompleted proteins in an environment suitable for their maturation, to favor their assembly, to inhibit protein aggregation and to maintain homeostasis in the early secretory pathway. In this scenario it is clear that the first step for the proteins to travel along the secretory pathway is to be mature, folded and quality control approved. This indicates that the folding process and quality control systems go hand by hand. Different genetic and environmental conditions including hypoxia, glucose deprivation and pH unbalances, may result improper protein folding, consequently creating a condition known as ER stress [21]. Mammalian ER stress response reacts to mis/malfolded protein insults in different ways including i) attenuating translation, ii) promoting expression of chaperones, iii) enhancing protein degradation component (ERAD) and/or iv) inducing apoptosis [22]. All these mechanisms are mediated by three ER transmembrane receptors (PERK, ATF6alpha and IRE1alpha) [23] that sense and monitor the protein folding status of the ER through their luminal domains. Upon activation these sensors activate a transcriptional program referred to as unfolded protein response (UPR). More recently, Golgi stress signaling pathways were also discovered but their level of characterization is much less (Figure 2).

#### 2.3.1. Translation Attenuation and mRNA Degradation

PERK is a type I transmembrane protein located in the ER membrane, it contains a luminal stress sensing domain that senses unfolded proteins accumulation in the ER and a cytosolic portion, which contains the kinase domain that transduces its activation [24,25]. In unstressed conditions PERK luminal domain is bound to the chaperone BiP, which inhibits its activity. Upon ER stress, BiP binds misfolded proteins and is released from PERK luminal domain allowing its activation through oligomerization and trans-phosphorylation. Activated PERK phosphorylates and inactivates the alpha-subunit of eukaryotic translational initiation factor 2 (eIF2alpha), leading to global translational attenuation [26]. The phosphorylation of PERK is transient as the protein is dephosphorylated by specific phosphatases such as CREP (constitutive repressor of eIF2alpha phosphorylation), protein phosphatase 2C-GADD34 and p58IPK [27]. CREP is constitutively expressed, whereas the expression of GADD34 and p58IPK is induced upon ER stress by PERK and activating transcription factor 6 (ATF6) pathways, respectively [27]. In parallel, to global mRNA translation inhibition, cells employ an additive pathway referred to as RIDD [28], which depends on IRE1lapha RNase domain. In particular, upon unfolded protein accumulation IRE1alpha receptor dimerize, followed by auto-transphosphorylation and oligomerization. During RIDD, IRE1alpha cleaves mRNAs encoding for ER-targeted proteins with harbors a signal peptide or transmembrane domains and that are translated by ribosomes on the ER membrane. In mammalian cells, RIDD targets are enriched with mRNAs containing a cleavage site with a consensus sequence (CTGCAG) and a predicted secondary structure similar to the conserved IRE1 recognition stem-loop of the XBP1 mRNA [28].

#### 2.3.2. Expression of Chaperones and ERAD Components

Chaperones and ERAD components are targets of the active portion of the UPR sensor ATF6alpha (hereafter referred to ATF6) [29]. ATF6 is a type II transmembrane protein, the cytosolic portion of ATF6 has a DNA-binding domain containing the basic-leucine zipper motif (bZIP) and a transcriptional activation domain. To be active, ATF6 is transported from the ER to Golgi [30]. There, ATF6 is sequentially cleaved by a pair of processing proteases called site 1 protease (S1P) and site 2 protease (S2P), and the resultant cytoplasmic portion of ATF6 (ATF6-f) translocates into the nucleus. ATF6-f binding site are three (ERSEI, ERSEII and unfolded protein response element UPRE) [31,32,33]. Genes such as BiP, GRP94 and calreticulin, which show these elements in their promoter, are ATF6-f targets [34]. The ERSEI consensus sequence is CCAAT-(N9)-CCACG, where ATF6 binds to the CCACG portion, whereas a general transcription factor, NF-Y (nuclear factor Y), binds to the CCAAT portion. In parallel to ATF6, the IRE1alpha target, XBP1s, also control chaperones and ERAD components production [35,36]. XBP1s mRNA is the product of the unconventional splicing mediated by IRE1alpha together with the tRNA ligase RtcB [37]. Diverse ERAD components are XBP1s and ATF6 targets including: EDEMs, HRD1, DERLIN-2,3, BiP, p58IPK, ERDJ4, PDI-P5 and HEDJ [34]. Interestingly, ATF6′s activation was found to be linked with diverse quality control proteins, such as ERp18, PDIA5 and EDEM1, [38,39,40] and with its glycosylation status [41]. In addition, it was shown that an IRE1alpha paralogue, named IRE1beta, could catalyze, to a lesser extent, the non-conventional splicing of XBP1 mRNA [42]. This data suggest that other parallel and non-canonical pathways can control UPR signals. In parallel to IRE1beta, an ATF6 paralog exists, ATF6beta, which has the same mechanism of activation of ATF6 [43]. Notably, ATF6beta was also found to bind the ERSE sequences together with NFY and mice lacking both ATF6 and ATF6beta die during embryogenesis [43] In addition to this, currently we know that at least five other ER-resident proteins share the same activation mechanism (CREBH, LUMAN, OASIS, BBF2H7 and CREB4) [44]. These proteins are activated by ER/Golgi stress and can regulate UPR target genes expression. However, they can be activated also in the presence of other stress types, suggesting their lower ER-stress specificity. For instance, ER stress strongly induces CREBH transcription, but its protein cleavage is modestly induced in the same conditions [44]. Notably, the cleaved portion was found to cooperate with ATF6 fragment in binding UPRE and ERSE reports. However, both protein and mRNA levels are upregulated upon inflammation and target genes involved in inflammatory response were found to be CREBH targets. Moreover, CREBH was demonstrated to be involved in transcription of genes involved in gluconeogenesis [45] and lipid metabolism in response to high fat diet. It would be interesting to understand which specific stimuli activates these ATF6-like transcription factor and whether in those condition ATF6 could help somehow in transcription and stress resolution.

#### 2.3.3. Mechanisms of Cell Death Induction

In conditions where ER stress become chronic and cannot be resolved, the UPR changes strategies from adaptive to destructive outputs through the induction of pro-apoptotic pathways. Many pathways are involved in ER stress-induced apoptosis and it seems to require the concomitant activation of several pathways to be induced. The main pro-apoptotic pathway is mediated by the transcription factor CHOP [46]. Its transcription could be dependent by both PERK and ATF6 upon ER stress. CHOP target genes are *GADD34* [47], *DR5* [48] and carbonic anhydrase VI, all of which appear to be involved in apoptosis. For instance, GADD34, which encodes for a subunit of protein phosphatase 1C, controls eIF2alpha dephosphorylation, thus promoting protein synthesis re-initiation. Notably, persistent protein synthesis, during ER stress, induces chronic UPR activation and consequently apoptosis. The other CHOP target, DR5, which encodes a cell surface death receptor that induces caspase activation cascades upon ligand engagement, was linked to ER-stress induced apoptosis in different cancer cell lines through unconventional activation mechanisms [48]. Finally, carbonic anhydrase VI contributes to ER stress-induced apoptosis, decreasing the intracellular pH [49]. In addition, the IRE1 arm of the UPR is involved in ER-stress induced apoptosis through to its kinase domain. The TRAF2/ASK1 complex, which promotes JNK phosphorylation and induces apoptosis, was found to be recruited to the IRE1alpha kinase domain [50]. Moreover, BCL2 family components appear to be active in ER stress and participate to the apoptotic pathway [23]. To this the anti-apoptotic factor BCL2 is downregulated by CHOP, leading to enhanced oxidant injury and apoptosis. The pro-apoptotic, BAX and BAK are active upon ER stress and associate with IRE1 can modulate its activity [51].

#### 2.3.4. Golgi and Mitochondrial Stress

All the organelle involved in the protein quality control are sensitive of unfolded protein accumulation and respond to this stress using different strategies. The Golgi and the mitochondria are prone to these stresses and are of interest in many recent studies, mostly because of the pivotal interconnection between all these organelles in maintaining cellular homeostasis [52,53]. Despite that, the molecular pathways involved in the stress responses of those organelles are poorly studied compared to ER stress. The peculiarity of the Golgi stress is that it is an autoregulated process in which three proteins were found to be important: TFE3, HSP47 and CREB3 [53]. TFE3 regulates genes necessary for increasing Golgi function after specific induction of Golgi stress (monesin treatment or SLC35A1 ablation) those genes include fucosyltrasferases and Golgi structural proteins [54]. In addition to this, monesin treatments were shown to induce expression of an ER-resident chaperone HSP47, which is not induced by ER stress [54]. Notably, HSP47 depletion not only induces Golgi fragmentation but also caspase3 and apoptosis induction, thus suggesting a pro-homeostatic role in Golgi stress [55]. Interestingly, HSP47 was shown to interact with IRE1alpha thus increasing its activation [56]. These data support the hypothesis of a strong connection between the two pathways and organelles. Additionally, Golgi stress induces apoptosis by activating CREB transcriptional activity. Upon Golgi stress, CREB3 is translocated from the ER to the Golgi, cleaved and transported as active transcription factor in the nucleus, where it transcribes for ARF4 and induces an apoptotic response in similar fashion to ATF6 [57]. In addition, CREB3L3 was shown to interact with ATF6-f and together induces expression of genes linked with gluconeogenesis [58].

### 2.4. ER Associated Degradation (ERAD), ER Reflux and Endosome and Golgi-Associated Degradation (EGAD)

Proteins that fail to comply with the ER quality control are then diverted to the cytosol where they are ubiquitinated and degraded by the proteasome in a process termed ER associated protein degradation (ERAD; Figure 3). ERAD-mediated process consists of four steps: recognition, translocation, ubiquitination and elimination. Proteins that are folded in the CANX/CALR cycle are recognized by three ER degradation-enhancing alpha-mannosidase-like proteins (EDEM1–3) that contain a mannosidase-like domain, which is responsible for the recognition of the mannose residues. EDEM1 is a transmembrane protein on the ER membrane that recognizes and extracts misfolded glycoproteins from the CANX/CALR pathway in N-glycan-independent manner. EDEM2 and EDEM3 are ER luminal proteins, EDEM2 recognizes and targets misfolded glycoproteins in an N-glycan-dependent manner while EDEM3 increases the degradation of glycoproteins by ERAD through trimming of the mannose from Man8GlcNAc2 to Man7GlcNAc2 [59]. Recently, the mechanism and the role of two other redundant ER lectins OS9 and XTP3B in the degradation of glycoproteins were reported [60]. Both lectins were found to be essential for efficient glycoprotein degradation and stabilization of the SEL1L/HRD1 dislocation complex, while OS9 binds both glycosylated and non-glycosylated proteins, XTP3B inhibits ERAD of non-glycosylated proteins that is antagonized by OS9 [61]. Once misfolded proteins are recognized they are usually associated with two important components, the protein disulfide isomerase (PDI) that is responsible for isomerization of disulfide bonds and BIP, which is needed in unfolding of the partially folded structure before translocation. To reach the cytoplasm, ER-resident proteins need to pass by the ER membrane, this process is mediated by the retrotranslocon. Historically, this structure was considered to be mainly composed of SEC61, the major component of the translocation channel that imports polypeptides into the ER; despite that the molecular structure of the retrotranslocon machinery still remains to be defined. Derlin proteins (Derlin-1, 2 and 3) are another family of proteins that participate in ERAD and part of the translocon component [2].

In particular, DERLIN-1 interacts with P97/VCP and VCP-interacting membrane protein 1 (VIMP1), this complex was suggested to act as the retro-translocation channel in the ER membrane [62]. The P97/VCP complex is necessary for the recruitment and the transport of unfolded protein to the cytosol, due to the help of the ubiquitin fusion degradation protein 1 (UFD1) and nuclear protein localization 4 (NPL4). The third step of ubiquitination is mediated by the ubiquitin-activating enzyme (E1)–ubiquitin conjugating enzyme (E2)–ubiquitin-protein ligase (E3) of the ubiquitin system [7]. Mechanistically, the ubiquitin, firstly, need to be conjugated to E2 enzyme by the E1 enzyme and then transferred to the substrates by E3 enzyme. HRD1, gp78 and TEB4/Doa10 belong to the E3 ligases class, whereas UBC6 and UBE2-G2/UBC7 are E2 ligases mainly involved in ERAD. UBE1 is an E1 ubiquitin-activating enzyme that controls polypeptides degradation by the proteasome through the formation of a thioester linkage between a cysteine residue in the E1 active site and the C-terminus of ubiquitin, a process that needs ATP hydrolysis. Notably, HRD1 preferentially targets misfolded luminal domains (ERAD-L) and DOA10 targets misfolded cytosolic domains (ERAD-C). The HRD1 complex is composed of HRD1, HRD3, DERLIN1, OS9, USA1, UBX2 and p97 (ERAD-L) [63]; Doa10 complex consists of DOA10, UBX2 and P97/VCP (ERAD-C). Interestingly, misfolded transmembrane domains bypass OS9 and HRD3 binding and are directly linked to the HRD1 complex (ERAD-M). Nonetheless, other E3 ligases exist that show different specificity for substrates [64]. Similar to ERAD, an endosome and Golgi-related stress-responsive associated degradation pathway (EGAD) exists in the Golgi [4]. EGAD is implicated in several processes including changes in the size, subcellular localization and organization of the Golgi, thus impacting on cell fate and cellular pathways. For instance, recently was demonstrated that Golgi stress may promote cells death in MEK1/2 and ERK1/2 dependent fashion [65,66]. Alternatively, it was also proposed that Golgi stress response confers cytoprotection in a neurodegenerative disease reprogramming cysteine metabolism [67]. EGAD mediated machinery was identified in *Saccharomyces cerevisiae* and was proposed to be mediated by the DSC complex [68]. This complex, similar to the ERAD HSD1 E4-ligase complex, is composed by six components: the membrane proteins TUL1, DSC2, DSC3, DSC4 and UBX3, and the cytosolic AAA+ ATPase Cdc48 (the homologous for the mammalian p97/VCP). Interestingly, similarly to ERAD, CDC48 mediates proteins membranes extraction and consequently proteasome degradation at the level of the Golgi compartment. ORM2, an ER-resident protein, represent one EGAD substrate and it is involved in sphingolipid biosynthesis, became poly-ubiquitinated in the membrane and then extracted by P97/VCP for degradation [69]. In this model, EGAD controls sphingolipid biosynthesis by preventing ORM2 accumulation thus contributing on lipid homeostasis in eukaryotic cells. Interestingly, the paralogue of ORM2, ORM1, does not represent an EGAD substrate. However, diverse DSC substrates have been identified, and their future characterization will bring new insight into EGAD [22].

### 2.5. Late Stages of ER-Dependent Membrane Transport

Once quality controlled, newly synthesized proteins need to reach ERES where they incorporate into carriers that are coated into the membrane-coating coat protein II (COPII) [70]. COPII complex, a set of highly conserved proteins, responsible for creating small membrane vesicles that originate from the ER and are enriched on ERES. In particular, Sar1, Sec23, Sec24, Sec13 and Sec31 are the main components of these vesicles. Sec23 and Sec24 form heterodimer and represent the inner COPII coat, whereas Sec13 and Sec31 hetero-tetramers and compose the outer COPII coat. Sar1 is a small GTPase component that initiates the formation of transport vesicles from the ER and for this it is recruited to the ER membrane [71]. The formation and movement of these COPII-derived vesicles is a crucial first step in the cellular secretion pathway, through which membrane and luminal cargo proteins are transported from their site of synthesis, at the ER, to other membrane compartments in the cell. Specific transporter molecules mediate the exit from the ER of certain glycoproteins, concentrating them into forward transport vesicles. One of the most important of these cargoes is ERGIC-53, which by binding high-mannose glycoproteins in the ER mediates their release in a calcium and pH-dependent manner [71]. Vesicle coat proteins could be also retro-transported from the cis-Golgi to the ER exists in a COPI vesicles dependent manner. The COPI coat is composed of seven different protein subunits to form an heptameric COP complex (α, β, β’, γ, δ, ε and ζ) [72] COPII- and COPI related transports control is necessary for cellular homeostasis maintenance and both are linked to ER/Golgi stress and UPR activation regulation. For instance, it was demonstrated that almost all the COPII components are XBP1s transcriptional target genes, thus suggesting that UPR effectors control cargo vesicles composition [73]. Indeed, IRE1alpha overexpression was demonstrated to be sufficient to suppress the defect linked to mutations in SEC genes [74]. Moreover, it was also proposed that the IRE1/XBP1s axis activated upon nutrient imbalance, mediates COPII dependent lipoprotein traffic, thus reversing liver steatosis and hypolipidemia and linking COPII vesicles formation to nutrient deprivation stress [75,76].

Interestingly, it was also proposed that CREB3L2, a Golgi stress mediator that shares the same activation mechanism as ATF6, is responsible for transcription of Sec23A and Sec24D, two components of the coat protein complex II (COPII), during the differentiation of hepatic stellate cells (HSCs) [77]. Moreover, CREB3L2 in medaka fish embryos appeared to control the transcription of all genes necessary for enlargement of COPII vesicles to improve collagen export from the ER [78]. Taken together these evidences all support the hypothesis that COPII formation is not only necessary for transport of folded proteins, but it is also controlled by ER stress, thus conditions that perturb proteostasis could be also linked to impaired COPII vesicles formation and all the secretory process.

## 3. Proteostasis and Secretory Pathway Related Diseases

Protein quality control is a stringent mechanism particularly for production and consequently secretion of mature proteins. Proteins that escape the quality control (QC) systems can aggregate both inside and outside the cells contributing to several human diseases. 

### 3.1. Protein Quality Control and Aggregation

Malfolded proteins are linked to ER/Golgi stress activation, protein quality control dysregulation and diverse human pathologies. For instance, neurodegenerative diseases, such as Parkinson’s disease (PD), Huntington’s disease (HD), Alzheimer’s disease (AD) and neuropathies (Charcot–Marie tooth (CMT), are related to gene mutations and consequently to production of immature proteins, prone to aggregate and to induce neurons cell death [79]. Immature proteins aggregates are able to induce both cellular stresses and to avoid quality control systems. For instance, all the aggregates linked to AD and PD were found to induce ER stress activation, but also to impinge on ER stress receptors activation, thus reducing the pro-homeostatic responses [80]. Interestingly, ERAD components mutation was associated with AD patients. The Z-variant of the alpha 1-antitrypsin (A1AT) was shown to activate ERAD-degradation, autophagy and ATF6 transcriptional activity [81,82,83]. Moreover, activation of the ATF6 pathway was also shown to attenuate Z-A1AT accumulation and mitochondrial damage in liver cells through promoting ERAD. PTMs were also associated with protein aggregates accumulations, for instance, phosphorylation, SUMOylation and ubiquitination have been shown to affect PD-related proteins, phosphorylation of a-synuclein in ser-129 increase aggregates formation and consequently toxicity [84] or hyperphosphorylated Tau have been detected in brains of patients with refractory epilepsy and AD patients [85]. Huntington disease appears to be linked with Golgi stress, since it was demonstrated to regulate cysteine metabolism through ATF4 activation [86]. Moreover, fragmented Golgi was observed in cell culture transfected with mutated version of Tau, compared to wild type [87].

### 3.2. Protein Quality Control and Degradation

A second class of proteostasis-related diseases are due to a more stringent quality control system that degrades proteins even if they are correctly folded. These diseases are considered loss-of function diseases and lysosomal storage diseases belong to this class. A more stringent ER quality control, usually, identifies misfolded or destabilized glucosyl ceramidase (GCase) variants thus promoting their degradation [79]. It was demonstrated that knockdown of IRE1alpha, PERK and ER proteostasis network components (FKBP10 and ERDJ3), which are responsible for GCase degradation, serve to enhance trafficking of a specific GCase variant, thus reducing its degradation [88]. Cystic fibrosis belongs to the same diseases family and was linked to increased ERAD activity. In particular, Cystic Fibrosis Transmembrane Conductance Regulator (CFTR) variants are ERAD targets, recognized by HSP70, HSP40 and small HSPs, polyubiquitinated and, finally, delivered to the proteasome, thus inducing their degradation [89,90]. An opposite effect has been shown for variants of transthyretin (TTR), which are able to avoid ERAD dependent quality control, thanks to ERDJ3-transport, thus forming amyloids in the heart and peripheral nerves [8,88]. Golgi fragmentation seems to be important for diverse loss of function diseases, for instance: the Smith-McCort Dysplasia, linked to reduced Ras-Related Protein RAB33B expression levels, or the Duchenne muscular dystrophy where the dystrophin (DMD) protein is not expressed leading to aberrant Golgi organization [91].

### 3.3. Increased Protein Production/Imbalance Demand vs. Capacity 

UPR signaling has been shown to be essential in maintaining glucose homeostasis, to this impaired ER functionality was also connected to many metabolic diseases, such as: obesity, brain and renal ischemia [92]. Interestingly, PERK and IRE1 pathways and their downstream effectors appear to be involved in those pathologies. Other metabolic diseases linked to increase accumulation of cholesterol, homocysteine accumulation and Ca^2+^ imbalance, such as atherosclerosis, are linked to ER stress induction and increases chaperones expression [92]. In this context, Golgi and, particularly the Conserved oligomeric Golgi complex (COG) complex, were shown to be necessary in Low-Density Lipoprotein (LDL) Receptor (LDLR) trafficking and lipid maturation, through the COPI vesicles [18]. Recently, ATF6 was proposed to play a central role in renal and cerebral ischemia [93]. In particular, ATF6 targets genes, which are mainly involved in proteostasis regulation, were linked to resistance to ischemia representing a new putative proteostasis-based therapeutic strategy [94]. It also known that glucose uptake is a feature for cancer cells, to this UPR role in this context was also explored, and a lot of works highlighted the importance of all the ER stress receptors and quality control effector in cancer progression [95]. Notably, tumors cells have developed diverse strategies to avoid apoptotic response and to take advantages from the pro-homeostatic pathways. As matter of fact, cancer cells were shown to have high ERAD-component expression levels together with increased proteins glycosylation and ER lumen dimension [7]. It is interesting to note that there is a tight interconnection between the different components, for instance, in hepatocellular carcinoma progression, P97/VCP differentially regulates IRE1alpha and ATF6 resulting in different outputs [96]. Moreover, P97/VCP and GCN2 were connected with cancer cell metabolism and proteostasis upon nutrient deprivation conditions [97]. Together with high glucose demand, tumor cells are always exposed to hypoxia and increased ROS production. To cope with hypoxic condition, cancer cells were shown to promote transcription of ER-oxidases (ERO1), trough PERK and ERO1 was correlated with poor prognosis in breast cancer patients [47,98].

## 4. Therapeutic Strategies to Control the Homeostasis of the Early Secretory Pathway

Disturbance of protein quality control systems, the ER/Golgi functions or protein secretion is considered as a strong cause of many human diseases associated with defects in the secretory pathways. Proteostasis-associated diseases differ from each other by different gene mutations, and the process and the machinery involved in their currency and progression. On the other hand, they all have one common feature, which is the accumulation of immature and/or mis/malfolded proteins (Figure 4). Here we highlighted diverse strategies that could be translated into the clinic to target the secretory pathway. Notably, since the strong connection between the different components, one strategy could be applied to contemporary hit diverse targets.

### 4.1. Proteostasis Modulation 

The use of compounds that attenuate stress by tempering protein misfolding is believed to be a promising strategy. As such tauroursodeoxycholate (TUDCA) [99], is a biliary salt that was found to reduce ER stress, to inhibit inflammatory response (NF-kB inhibitors) and to act as a cytoprotective especially in pancreatic beta cells and hepatocytes. TUDCA works mainly to reduce ROS and decrease mitochondria dysfunction in addition to reduce protein aggregates in the cell. These characteristics place TUDCA as a promising candidate for ER stress mediated diseases and in fact TUDCA is now translated in several clinical trials studies (TTR amyloidosis or liver cirrhosis) [99,100]. In addition, 4-phenylbutyrate (4-PBA) is an FDA (Food and Drug Administration) approved compound to treat urea acid disorders. 4-PBA is a histone deacetylase (HDAC) inhibitor whose activity was successfully demonstrated in several models of human diseases. For instance, administration of 4-PBA has been shown to effectively stabilize mutant CFTR (ΔF508-CFTR) and decrease its degradation resulting in its increase plasma membrane localization [101]. Another histone deacetylase inhibitor SAHA was also shown to increase ΔF508 CFTR glycosylation, thus partially rescuing its function [102]. Oral administration of TUDCA or 4-PBA was shown to inhibit hyperglycemia, improve glucose tolerance, decreased stress inside the ER in response to mis/malfolded protein production and increase insulin receptor signaling. Moreover, 4-PBA has also been shown to increase the secretion of the mutant A1AT protein, while TUDCA inhibits apoptosis induced by mutant A1AT protein and reduces hepatocellular carcinogenesis in an HCC model [103].

### 4.2. Gene Therapy

Gene therapy is becoming increasingly important in diseases characterized by single gene mutations. One example is the treatment of the Gaucher disease, in which the wild-type (wt) variant of the protein is introduced to correct an existing mutant protein through intravenous administration. However, low amounts of the injected enzyme to reach the target organelle, reducing the efficacy of the treatment. A new delivery strategy was developed through the use of the adeno-associated viral (AAV) vector, which showed promising results in clinical trials for several diseases including monogenic disorders. Moreover, administration of adeno-associated viral vectors encoding the ER chaperone BiP in the Substantia Nigra pars compacta (SNpc) was associated with diminished dopaminergic neuron loss induced by αSYN overexpression [104,105]. Delivery of the αSYN target RAB1, using AAVs, has been tested as a strategy to restore protein maturation in PD models [106]. Alternatively, lentivirus-mediated protein delivery, was shown to protect from locomotor deficits induced by αSYN overexpression [107]. PD development was also associated with low XBP1 activity, a matter of fact XBP1 depletion was increasing ER stress and apoptosis in adult SNpc, to this gene therapy to deliver an active form of XBP1 was used [105]. Interestingly, neuroprotection and reduced striatal denervation was observed in animals injected with 6-hydroxydopamine [108]. The same interesting results were obtained in Huntington’s disease models. Here, AAV-mediated delivery of XBP1s was sufficient to reduce mutant huntingtin levels in mice models of these diseases, suggesting that gene therapy could represent a promising strategy to improve protein folding in proteinopathies [109].

### 4.3. Targeting Protein Folding and Quality Control Machineries

Most of the components of the secretory and protein quality control pathways could potentially target small molecules proteostasis regulators. The ERQC for glycoproteins can be altered by α-glycosidase inhibitors CST (dual ER α-Glu I/II), 4-dideoxy-1,4-imino-D-arabinitol (DAB, ER α-Glu II) and 2,5-dideoxy-2,5-imino-D-mannitol (DMDP, ER α-Glu I) and the mannosidase inhibitor kifunensine (KIF, ER Man I) [110] are used in the clinic together with iminosugars [111] a class of α-glycosidases inhibitor, which mimic carbohydrate structure preventing N-glycan trimming. For their inhibitory activity, iminosugars are used in the treatment of various diseases, such as cancer, viral infections and cystic fibrosis [111]. P97/VCP is considered to be upregulated in multiple diseases and cancers, this highlights its importance as a potential therapeutic target. DBeQ, ML240, ML241 and CB-5083, selective and potent P97/VCP inhibitors, rapidly cause cancer cell death [62]. CB-5083 selectively binds P97/VCP D2 domain and induce proteotoxic stress through CCAAT/enhancer-binding protein homologous protein (CHOP) and death receptor 5 (DR5) upregulation [112,113]. In 2017, preclinical data for CB-5083 later showed efficacy in several multiple myeloma (MM) models [114]. NMS-873, an allosteric inhibitor of P97/VCP, was shown to interfere with protein degradation and to induce cancer cell death [96]. The cargo receptors also were shown to be pivotal for proteinopathies. Recently, a molecule called BRD4780 was shown to be able to remove MUC-1 mutant misfolded protein from the cargo receptor TMED9 [96]. This retention mechanism is at the basis of an autosomal kidney condition known as MKD. Interestingly, it was also shown that BRD4780 mediates the clearance of others misfolded proteins, suggesting that it could be applied for different proteinopathies [115]. PDI proteins family, such as ERP57 or ERP72, involved in the onset of different neurodegenerative diseases and upregulated in cancer, could be targeted using PACMA31 [116]. This molecule was shown to bind the active-site cysteines in PDIs, thus inhibiting their activity and inducing cytotoxicity in ovarian cancer cells. In the other hand, the 16F16 compound is employed to increase PDIs activity and to suppress apoptosis in HD cell lines model [117,118]. Finally, a series of sulfhydryl reagents were reported to inhibit the catalytic activity of PDI, by reacting with the free thiol groups in PDI and therefore, acting as irreversible inhibitors [116]. Some small molecules block later stages in the secretory pathway such as Golgi transport. One such molecule is the Golgicide A (GA), which induces the dislocation of markers of the cis-Golgi without affecting morphology of the ERGIC and endocytic pathways. Clinically, it was shown to inhibit protein secretion, arrest Shiga toxin trafficking in the endosomal compartment and disrupt epithelial apical junctions [119].

### 4.4. Targeting the UPR

Compounds interacting directly with UPR mediators (e.g., PERK, IRE1a or ATF6) can also be valuable therapeutic agents for the treatment of neurodegeneration and cancers. For instance, the PERK/eiF2a pathways are crucial in the onset of prion like diseases, different cancers, metabolic diseases and other disease models. Targeting of the PERK/eIF2α pathway using the potent PERK inhibitor GSK2606414 [119,120] resulted in neuroprotection in prion disease and Tau-mediated frontotemporal dementia models. Oral administration of GSK2606414 resulted in effective neuroprotection, preventing nigral-dopaminergic neuron loss in mice treated with a PD-inducing neurotoxin [116]. These positive effects of GSK2606414 administration were accompanied by augmented dopamine levels in the striatum, improved motor performance and increased expression of synaptic proteins. Thus, inhibiting PERK signaling could be considered as an interesting candidate for disease intervention. However, GSK2606414 administration has resulted in severe side effects related with pancreatic toxicity, in addition to off-target effects on RIP kinases. Persistent activation of eIF2α has been linked to the development of several neurological disorders and its modulation could be promising as therapeutic strategy. For example, the small molecule Salubrinal^®^ inhibits eIF2α phosphatase complexes, GADD34-PP1 and the constitutive complex (CREP-PP1) [121]. At last, ISRIB was identified as inhibitor eIF2B and was shown to increase cognitive memory process in mice, to be protective in mouse models of neurodegeneration and in traumatic brain injury [122]. Notably, it resulted less toxic compared to the PERK inhibitor when injected in mice. Moreover, as IRE1alpha was linked to different aspects of proteinopathies, the use of IRE1 modulators could become of interest. For instance, kinase-inhibiting RNase attenuators (KIRAs) were developed to inhibit the IRE1 arm of the UPR. KIRA6 and KIRA8 were shown recently to inhibit IRE1a in vivo and promote cell survival during ER-stress conditions [123,124]. In addition, other compounds that target the IRE1 endoribonuclease domain, such as toyocamycin, STF-083010, 4µ8C, MKC-3946 and B-I09, are also highly used in different diseases models. STF-083010, MKC-3946 and toyocamycin demonstrated therapeutic efficacy in multiple myeloma xenograft models [125,126,127], and B-I09 has been shown to control the aggressiveness of chronic lymphocytic leukemia cells in vivo [128]. Salubrinal^®^, a registered drug compound, was mainly reported to directly inhibit the GADD34-phosphatase complex that dephosphorylates eIF2a (PERK-cascade). Thus, Salubrinal^®^ was demonstrated to be a cytoprotective agent against ER stress [121,129]. Finally, ATF6 pro-homeostatic activity was linked to all those pathologies in which proteostasis results impaired. In this context, very recently, an activator of ATF6 was tested and it was found as protective in several pathologies such as renal/cerebral ischemia and proteins aggregation-related diseases. In particular, it was show that the compound 147 induces ATF6 by selectively modifying ER resident proteins that regulate ATF6 activity, such as PDIs, which suggest the tight connection between the different components of the ER stress responses [130]. In parallel, ATF6 inhibitors are employed against cancer, for instance Nelfinavir, firstly identified as viral proteases inhibitor, was then shown as S1P/S2P proteases inhibitor together with its derivate. This compound was show increase cell death in prostate and breast cancer cells [131,132]. However, giving the non-specific action of Nelfinavir, another class of ATF6 inhibitors has been recently discovered in the Walter lab. Ceapins, a new class of pyrazoleamides, have been recently demonstrated to specifically inhibit the ATF6alpha branch of the UPR by blocking ATF6a processing and nuclear translocation in cells undergoing ER stress. Ceapins are rather toxic, therefore further optimization of Ceapins for in vivo use should be done [133,134].

## 5. Future Perspectives

An even more increasing number of diseases are associated with the accelerated production and/or delayed clearance of protein aggregates, which point the attention on developing new drugs as a crucial objective. Studying the molecular mechanisms on the basis of ER stress and proteostasis has allowed scientists to discover the existence of a strong connection with various diseases. From a clinical point of view, these discoveries were essential to uncover new therapeutic targets and approaches to struggle diverse pathologies. The diversity between the diseases make the duty difficult, however lingering on the similarities, for instance between the neurodegenerative diseases, we could observe neuronal loss and proteins aggregation in the brain. This aspect should be taken in account, in a clinical prospective, pointing the attention on founding new apoptotic factors inhibitors or new molecular chaperones that could reduce aggregates accumulation. Some compounds have already been identified but an effort to find new possibilities for patients with cystic fibrosis and α1-AT deficiency should be done. For the amyloid-deposition diseases, a strategy could be the stabilization of the mature protein or the depletion of the precursors. Activation of the immune system against these aggregates, using a kind of vaccine, could be a promising strategy. Important progresses were made also in understanding the role of UPR and proteostasis in oncogenesis and cancer progression. In this context, ER stress and proteostasis are protective and the goal is to increase immature proteins accumulation to overcome the ability of the cells to survive. In this prospective, drug screenings were made to identified compounds that could be useful to target proteostasis and its pro-homeostatic effectors together with anti-tumor effects. To this, combinations of UPR inhibitors with conventional and non-conventional anti-cancer drugs could be a strategy. However, since in proteostasis and cancer development are involved also other processes, such as hypoxia, DNA damage and calcium balance, compounds that targets also these pathways could give higher effects compared to the single treatments. In addition, it is known that cancer cells are able to transmit signals to the surrounding cells in the microenvironment due to secretion of diverse molecules. For better understanding, deeper studies on secretion should be done. Therefore, it could be necessary to find a way to manipulate specific UPR or secretory pathway components to find single targets for particular tumors type. Development of less toxic and more bio-available versions of drugs could amelioration these molecules and may represent an optimal strategy to improve specificity of treatments.

## Figures and Tables

**Figure 1 cells-08-01347-f001:**
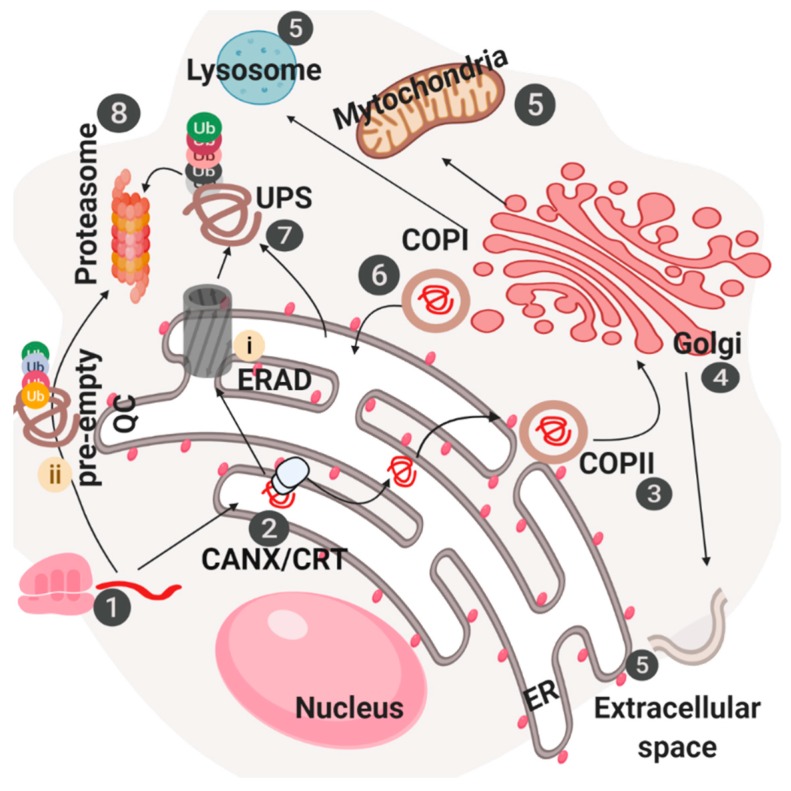
Molecular machines involved in the control of protein homeostasis in the early secretory pathway. After ribosomal-dependent messenger RNA (mRNA) translation (**1**), newly synthesized polypeptide chains are imported in the endoplasmic reticulum (ER) lumen. Here, proteins encounter diverse chaperones, modifying enzymes and complexes, such as CANX/CRT, necessary for maturation and folding (**2**). Once ready, mature proteins are embedded into COPII vesicles (**3**) and transported into the Golgi compartment (**4**). Here other chaperones and enzymes complete proteins maturation. Finally, once completely ready proteins are translocated in diverse subcellular compartments (extracellular space, mitochondria, lysosomes, etc.) (**5**). In the other hand, COPI vesicles mediate Golgi-to-ER retrograde transport of immature proteins (**6**). Proteins that are not fully mature and do not pass ER/Golgi dependent quality control systems are retrieved back in the cytosol through the ER-associated degradation (ERAD) machinery (**i**), targeted by ubiquitin system (**7**) and finally degraded by proteasome (**8**). Alternatively, newly synthetized polypeptide chains, in some conditions, do not reach the ER lumen and are directly ubiquitinated and degraded by proteasome, through a process referred to as pre-empty quality control (pre-QC) (**ii**).

**Figure 2 cells-08-01347-f002:**
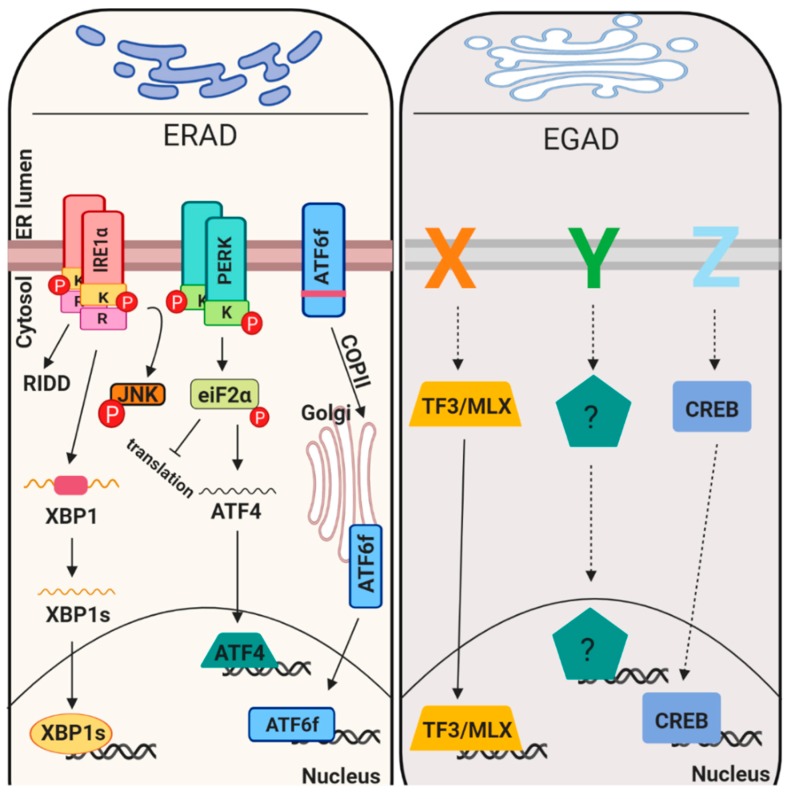
Organelle homeostasis imbalance signaling pathways. ER stress signaling (with the unfolded protein response (UPR) sensors IRE1, PERK and ATF6; left) and Golgi stress response (right). Left panel: unfolded protein accumulation is sense by three ER resident receptors: IRE1, PERK and ATF6. IRE1 activation imposes the induction of two different activities: kinasic and RNAsic activity. The first controls phosphorylation of the pro-apoptotic factor JNK, the latter mediates IRE1 dependent mRNA decay (RIDD) and the unconventional splicing of XBP1 mRNA. PERK activation induces eiF2a phosphorylation, thus inducing global mRNA translation inhibition and translation of specific mRNA, such as ATF4 mRNA. ATF6 activation induces its translocation into the Golgi, where it is the substrate of two proteases. This cleavage promotes the release of an active transcription factor known as ATF6f. All together these factors mediate the transcriptional program unfolded protein response. Right panel: unfolded protein accumulation also induces a Golgi-dependent response. The main receptors are still needed to be defined (X, Y and Z), but currently two main effectors were identified: TF3/MLX and CREB3. The first regulates genes necessary to increase Golgi function, the latter transcribe for ARF4, thus inducing apoptosis.

**Figure 3 cells-08-01347-f003:**
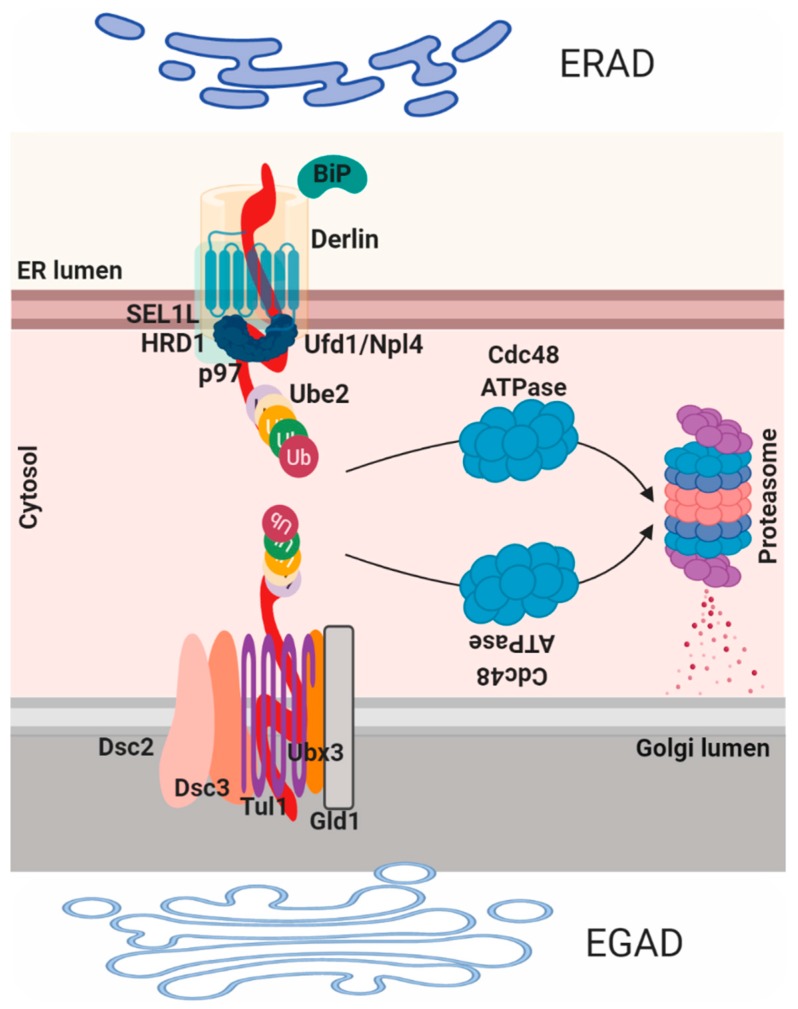
ER and Golgi degradation pathways, namely ERAD (upper panel) and endosome and Golgi-related stress-responsive associated degradation pathway (EGAD; lower panel). Upper panel: ERAD-mediated process consists of four steps: recognition, translocation, ubiquitination and elimination. To reach the cytoplasm, ER-resident proteins need to pass by the ER membrane, this process is mediated by the retrotranslocon. SEC61 is the major component of the translocation channel that imports polypeptides into the ER. Derlin proteins participate in ERAD and are part of the translocon component. Derlin-1 interacts with p97 and VCP-interacting membrane protein 1 (VIMP1) complex necessary for the recruitment and the transport of unfolded protein to the cytosol, thanks to the help of ubiquitin fusion degradation protein 1 (UFD1) and nuclear protein localization 4 (NPL4). UBE1 is an E1 ubiquitin-activating enzyme controls polypeptides degradation. The HRD1 complex is composed of HRD1, HRD3, Derlin1, OS9, USA1, UBX2 and p97 (ERAD-L); Doa10 complex consists of Doa10, UBX2 and p97 (ERAD-C). Lower panel: EGAD (endosome and Golgi-related stress-responsive associated degradation pathway) controls changes in the size, subcellular localization, and organization of the Golgi. The Dsc complex is composed of six components: the membrane proteins Tul1, Dsc2, Dsc3, Dsc4 and Ubx3, and the cytosolic AAA+ ATPase Cdc48 (the homologous for the mammalian p97/VCP).

**Figure 4 cells-08-01347-f004:**
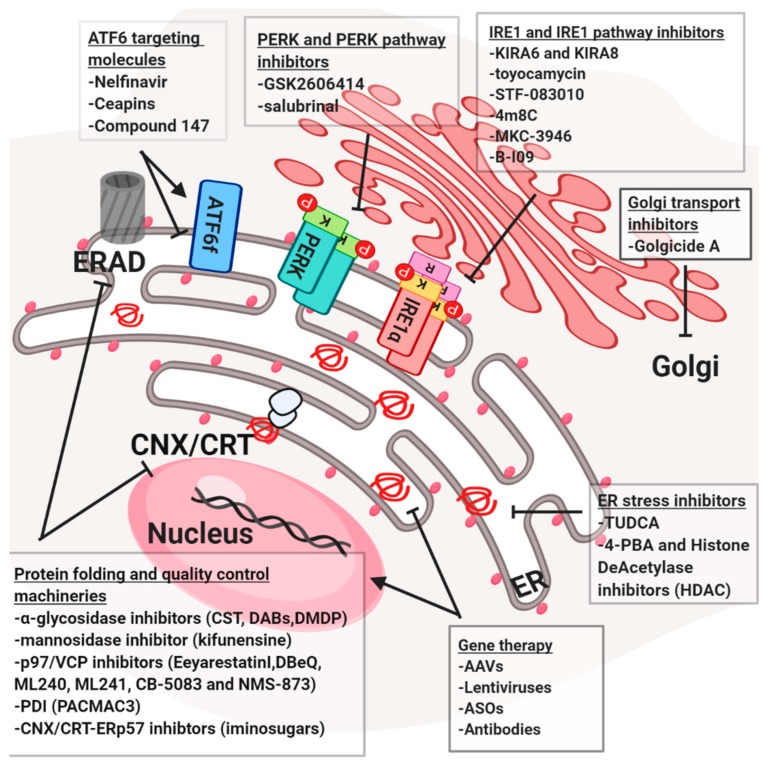
Targeting different steps in the early secretory pathway. Schematic presentation of proteostasis targeting compounds used in clinic for the treatment of diverse diseases.

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
