# Peer review of "Control of Protein Homeostasis in the Early Secretory Pathway: Current Status and Challenges"

_cells, 2019, doi:10.3390/cells8111347_

Round 1
Reviewer 1 Report
Sicari et al. have written a comprehensive review on the regulation of protein homeostasis in the early secretory pathway, and the contribution that dysregulation of this pathway can lead to a number of diseases. They discuss these diseases in detail and present strategies for therapeutically targeting the misfolded proteins that lead to disease pathogenesis. It is an interesting review and I recommend that it be accepted with the following minor revisions.
There are minor grammatical errors throughout. The manuscript should be proof-read again with a view to removing them. The text in the figures is too small to be easily read. Figure 3 Legend: should be Upper Panel and Lower Panel Should line 390 read “…..linked to AD and PD…..”? Figure 4 Legend: “…used in the clinic….”?
Reviewer 2 Report
In Sicari et al., the authors summarize how the endoplasmic reticulum and the Golgi, early stages of the secretory pathway, control the proper folding of proteins. In the first part of the of the manuscript, they described the mechanism to deal with the accumulation of improperly folded proteins and how the failure of these mechanism is linked to several human disorders. While, at the end of the manuscript, they mentioned the pharmacological and genetic strategies that might result in putative therapeutic tools to treat these conditions.
Although the manuscript mentioned all the different pathways involved in the control of protein homeostasis, there are some inadequacies along the text that made it confusing and hart to read.
The quality of the figures is quite poor (on screen or printed version), which makes difficult to read the labelling. This is particularly true for the figure 3 and 4, where the labels are illegible.
In the text there are some conceptual and grammar mistakes that should be addressed:
Page 2, line 68 “ATPase domains, among them ERdj3 , (missing) rely…in the ER. It binds (lack of consistency, it should plural…) BiP to its ATP bound form and is necessary for BiP to recruit its substrates” The whole paragraph is not clear at all. I figure that they are talking about the ATPase domain of HSP40
Page 2, line 72 “Moreover, ERdj6 was found together with, ERdj5 in association with ERAD…”??
Page 2 line 78 & 82 Use of abbreviations without mentioned before PDI and UPR. They are properly described in page 4, and they repeat them again in page 8
Page 2 line 79 “Bip”, (delete comma).
Page 3 line 95 “ to theMan9 glycan”
Page 7 line 241 “GADD34 associated with protein phosphatase 2C enhances dephosporylation of eIF2alpha and promotes*** DR5, which encodes for a cell surface death receptor and induce caspase cascades and carbonic anhydrase VI change the cellular PH.” ???
Page 7 line 263 Interesringly ,
Page 8 line 293 “still remains to be defined”
Page 9 line 390 “linked to AS”
Page 11 line 449 “of of immature”
Page 13 line 485 Delivery of the αSYN target, RAB1, using AAVs has been tested as a
486 strategy to restore protein maturation in PD models [108]. The expression delivery is confusing
Page 13 line 493 “levels .in”
Line 500 “.are”
Line 508 “MM models” MM stands for…?
Line 515 "PDI proteins, (which are ?) involved in onset of different neurodegenerative diseases and upregulated in cancer, could be targeted using (missing) [119]"
Line 539 pancreatic toxicity, in addition to off-targeting effects over the RIP kinases. Persistent activation of the has been eIF2α linked to the development of several neurological disorders and its modulation
